# The Effect of Cerium Oxide (CeO_2_) on Ischemia-Reperfusion Injury in Skeletal Muscle in Mice with Streptozocin-Induced Diabetes

**DOI:** 10.3390/medicina60050752

**Published:** 2024-04-30

**Authors:** Abdullah Özer, Necmiye Şengel, Ayşegül Küçük, Zeynep Yığman, Çağrı Özdemir, Yiğit Kılıç, Ali Doğan Dursun, Hasan Bostancı, Gülay Kip, Mustafa Arslan

**Affiliations:** 1Department of Cardiovascular Surgery, Faculty of Medicine, Gazi University, Ankara 06510, Turkey; dr-abdozer@hotmail.com; 2Department of Oral and Maxillofacial Surgery, Faculty of Dentistry, Gazi University, Ankara 06490, Turkey; necmiyesengel@hotmail.com; 3Department of Physiology, Faculty of Medicine, Kutahya Health Sciences University, Kutahya 43020, Turkey; aysegul.kucuk@ksbu.edu.tr; 4Department of Histology and Embryology, Faculty of Medicine, Gazi University, Ankara 06510, Turkey; zeynepyigman@gmail.com; 5Neuroscience and Neurotechnology Center of Excellence (NÖROM), Gazi University, Ankara 06830, Turkey; 6Department of Anesthesiology and Reanimation, Faculty of Medicine, Gazi University, Ankara 06510, Turkey; mdcagriozdemir@gmail.com (Ç.Ö.); gulaykip@yahoo.com (G.K.); 7Department of Pediatric Cardiovascular Surgery, Gazi Yaşargil Education Research Hospital, Diyarbakır 21010, Turkey; dr-yigit@yandex.com; 8Department of Physiology, Faculty of Medicine, Atılım University, Ankara 06830, Turkey; alidogandursun@gmail.com; 9Department of General Surgery, Faculty of Medicine, Gazi University, Ankara 06510, Turkey; hasanbostanci@gazi.edu.tr; 10Life Sciences Application and Research Center, Gazi University, Ankara 06830, Turkey; 11Laboratory Animal Breeding and Experimental Researches Center (GÜDAM), Gazi University, Ankara 06510, Turkey

**Keywords:** diabetes mellitus, cerium oxide, skeletal muscle, lower limb, ischemia-reperfusion

## Abstract

*Objective*: Lower extremity ischemia-reperfusion injury (IRI) may occur with trauma-related vascular injury and various vascular diseases, during the use of a tourniquet, in temporary clamping of the aorta in aortic surgery, or following acute or bilateral acute femoral artery occlusion. Mitochondrial dysfunction and increased basal oxidative stress in diabetes may cause an increase in the effects of increased reactive oxygen species (ROS) and mitochondrial dysfunction due to IRI. It is of great importance to examine therapeutic approaches that can minimize the effects of IRI, especially for patient groups under chronic oxidative stress such as DM. Cerium oxide (CeO_2_) nanoparticles mimic antioxidant enzymes and act as a catalyst that scavenges ROS. In this study, it was aimed to investigate whether CeO_2_ has protective effects on skeletal muscles in lower extremity IRI in mice with streptozocin-induced diabetes. *Methods*: A total of 38 Swiss albino mice were divided into six groups as follows: control group (group C, *n* = 6), diabetes group (group D, *n* = 8), diabetes–CeO_2_ (group DCO, *n* = 8), diabetes–ischemia/reperfusion (group DIR, *n* = 8), and diabetes–ischemia/reperfusion–CeO_2_ (group DIRCO, *n* = 8). The DCO and DIRCO groups were given doses of CeO_2_ of 0.5 mg/kg intraperitoneally 30 min before the IR procedure. A 120 min ischemia–120 min reperfusion period with 100% O_2_ was performed. At the end of the reperfusion period, muscle tissues were removed for histopathological and biochemical examinations. *Results*: Total antioxidant status (TAS) levels were found to be significantly lower in group DIR compared with group D (*p* = 0.047 and *p* = 0.022, respectively). In group DIRCO, total oxidant status (TOS) levels were found to be significantly higher than in group DIR (*p* < 0.001). The oxidative stress index (OSI) was found to be significantly lower in group DIR compared with group DCO (*p* < 0.001). Paraoxanase (PON) enzyme activity was found to be significantly increased in group DIR compared with group DCO (*p* < 0.001). The disorganization and degeneration score for muscle cells, inflammatory cell infiltration score, and total injury score in group DIRCO were found to be significantly lower than in group DIR (*p* = 0.002, *p* = 0.034, and *p* = 0.001, respectively). *Conclusions*: Our results confirm that CeO_2_, with its antioxidative properties, reduces skeletal muscle damage in lower extremity IRI in diabetic mice.

## 1. Introduction

Diabetes mellitus (DM) is a chronic disease that is common in the world and is rapidly increasing. Dealing with diabetes and its complications is becoming a serious problem for the global health and economy [1]. According to the estimates of the International Diabetes Federation (IDF), the number of diabetics, which was 382 million in 2013, will increase to 592 million in 2035. In 2040, it is estimated that one in 10 adults will have diabetes (642 million) and healthcare expenditures for diabetes-related diseases will exceed USD 802 million [2]. Various complications, especially microvascular and macrovascular complications, can be seen in DM. It is stated that in diabetes, endogenous ROS increases, antioxidant defense systems are depleted, and accordingly, basal oxidative stress is already high. As a result of all these, the effects of ischemia-reperfusion injury increase in diabetes [3].

Lower extremity ischemia-reperfusion injury can occur with vascular injuries due to trauma and various vascular diseases [4], and it can also be seen during vascular or orthopedic surgeries during vascular clamp or tourniquet applications [5] for clear vision in the surgical field. Ischemia occurs when arterial blood flow is blocked and blood flow to organs is restricted; as a result, cellular energy stocks are depleted, a serious imbalance occurs in cell metabolism, toxic metabolites accumulate, and cell death may occur. Reperfusion injury may also occur when blood flow is restored to the ischemic tissue, which may lead to excessive damage and excessive immune response in the ischemic tissue [6].

Ischemia causes hypoxia, dysfunction of the mitochondrial electron transport chain, decrease in ATP production, and Na^+^-K^+^-ATPase and Ca^2+^-ATPase pump dysfunction. A decrease in ATP production induces anaerobic metabolism in mitochondria and causes a decrease in ATP and antioxidative agents in the cell. Intracellular reactive oxygen species (ROS) increase, and increasing lactic acid after anaerobic metabolism causes metabolic acidosis. Na^+^-K^+^-ATPase pump dysfunction reduces Na^+^-H^+^ pump activity by causing Na^+^ retention inside the cell and K^+^ accumulation outside the cell. The intracellular pH decreases, and Ca^2+^-ATPase dysfunction also impairs Ca^2+^ reuptake in the endoplasmic reticulum. Increased intracellular H^+^, Ca^2+^, and Na^+^ cause hyperosmolarity. A decrease in pH affects enzymatic activities and protein synthesis in ribosomes is reduced. Increased ROS causes oxidative stress, endothelial dysfunction, DNA damage, and increased local inflammatory responses [7].

The increase in molecular oxygen in the tissue resulting from the restoration of blood flow to the tissue in reperfusion causes a further increase in ROS production resulting in the activation of multifactorial complex mechanisms such as Ca^2+^ overload, the development of endothelial dysfunction, an increase in the prothrombogenic appearance, and exacerbations. It is caused by the immune response. Importantly, this situation affects both the local tissues where ischemia occurs but can also affect distant organs through the release of mediators accumulated in the ischemic tissue into the systemic circulation by blood flow being restored to the ischemic tissue [8].

The development of new treatment approaches that can reduce the negative effects of IRI is of great importance, especially for patient groups under chronic oxidative stress such as DM. Recently, studies have focused on the antioxidant-enzyme-like effects [9] and anti-inflammatory properties [10] of nanoparticles. One of these nanoparticles is cerium oxide (CeO_2_).

Cerium is a member of the lanthanide group and is the most abundant rare-earth metal with atomic number 58, is found in two oxidation states, i.e., 3+ and 4+ [11]. With the highly reactive surface area provided by the fluorite crystal, the cerium oxide nanoparticle lattice structure helps neutralize free radicals [12]. CeO_2_ nanoparticles enter the cell via receptor-mediated endocytosis. Their behavior as an antioxidant or pro-oxidant varies depending on the cytoplasmic pH. In normal cells with physiological pH, CeO_2_ nanoparticles act as a catalyst that scavenges ROS or free radicals by acting as antioxidant enzymes such as catalase and superoxide dismutase (SOD). The basis of their antioxidant activity is based on the redox cycle between the 3+ and 4+ states on their surfaces, the more Ce^4+^ states there are, the higher the catalase activity, and the more Ce^3+^ states there are, the higher the superoxide dismutase (SOD) activity [13].

On the other hand, cancer cells have a more acidic cytoplasm than healthy cells due to increased glycolysis and lactic acid production. While this acidic pH increases the SOD activity of CeO_2_ (reduction of superoxide to H_2_O_2_), it decreases the catalase activity, causing H_2_O_2_ accumulation in the cell and ensuring the apoptosis of the cancer cell [14]. Moreover, CeO_2_’s pro-oxidant properties at acidic pH can be used to create a cytotoxic effect on bacteria [15]. CeO_2_ nanoparticles were shown not to have significant cytotoxicity in in vitro studies with mammalian cells; it was observed that they had protective effects [16].

Cerium oxide has positive effects on diabetes and cerium oxide nanoparticles reduce plasma sugar to a similar extent compared to metformin in streptozocin-induced diabetic rats, and improve pancreatic functions by reducing inflammation and increasing beta cell proliferation [17]. It has been shown in some studies in the literature that CeO_2_, with its antioxidant-enzyme-like effect, reduces various organ and tissue (lung, liver, testicles and myocardial tissue) damage caused by free oxygen radicals formed during ischemia-reperfusion periods [18,19,20,21].

Their wide range of uses, different behaviors, antioxidant activities, and ability to self-recycle the surfaces of CeO_2_ nanoparticles from the oxidized (4+) state to the reduced (3+) state, and thus, restore their antioxidant capacity, make them unique [22].

This study is one of the first studies to examine the effect of lower extremity ischemia-reperfusion injury on skeletal muscle in diabetic mice. With this study, we wanted to contribute to the limited literature by aiming to investigate whether CeO_2_ has protective effects on skeletal muscles in lower extremity IRI in mice with streptozocin-induced diabetes.

## 2. Materials and Methods

The present study was conducted at the Gazi University Animal Experiments Laboratory (Ankara, Turkey) in accordance with the ARRIVE guidelines. The study protocol was approved by the Animal Research Committee of Gazi University (G.Ü.ET-22.063). All of the animals were maintained in accordance with the recommendations of the National Institutes of Health Guidelines for the Care and Use of Laboratory Animals.

In this study, a total of 38 Swiss albino mice (20–25 g) were used. The mice were housed in the laboratory at 20–21 °C for 12 h in daylight and 12 h in darkness. Free access to food was allowed. For the experiment, fasting was provided for 2 h before the application of anesthesia. Each mouse was numbered and they were divided into five groups using the sealed envelope method: control group (group C, *n* = 6), diabetes group (group D, *n* = 8), diabetes–CeO_2_, (group DCO, *n* = 8), diabetes–ischemia/reperfusion (group DIR, *n* = 8), and diabetes–ischemia/reperfusion–CeO_2_ (group DIRCO, *n* = 8). The groups were set up in this way because the ethics committee allowed 38 animals for the study.

Diabetes was induced by a single injection of streptozotocin (STZ) (Sigma Chemical, St. Louis, MO, USA) administered at 125 mg/kg intraperitoneally (i.p). The mice with fasting blood glucose (FBG) values higher than 250 mg/dL 72 h after injection were classified as diabetic. Animals with FBG levels > 250 mg/dL were included in the diabetic groups (group D, D-CeO_2_, DIR-CeO_2_). The mice were kept alive for 4 weeks after the STZ injection to allow the development of chronic diabetes before the study. Before the experiment, all mice were anesthetized using 50 mg/kg of intramuscular ketamine (Ketalar^®^; 1 mL = 50 mg; Pfizer, Istanbul, Turkey) and 10 mg/kg of xylazine hydrochloride (Alfazyne^®^ 2%, Ege Vet, Izmir, Turkey). A heating lamp was used to prevent heat loss during the experiment. The procedures were performed with the mice in the supine position.

Group C and group D: After skin asepsis, a midline laparotomy was performed without additional surgical intervention.

Group DCO: CeO_2_ (Co aqueous nanoparticle dispersion, 100 mL; Sigma-Aldrich; Merck KGaA, Darmstadt, Germany) was administered i.p (0.5 mg/kg) 30 min before the surgical procedure.

Group DIR: A midline laparotomy was performed and an atraumatic microclamp was placed on the infrarenal abdominal aorta for 120 min, then the clamp was withdrawn and the lower extremity was reperfused for another 120 min.

Group DIR-CO: CeO_2_ was administered i.p 30 min before the IR procedure.

To avoid hypovolemia, an hourly 3 mL/kg i.p isotonic solution was administered. During the IR period, the abdomen was covered with a moistened sterile pad. At the end of the experiments, after the animals were euthanized, the left hind leg was dissected, the calf muscles were isolated, and the gastrocnemius muscle was dissected. Gastrocnemius tissue samples were taken for biochemical and histopathologic analyses. Based on previous studies, we applied 2 h of ischemia and 2 h of reperfusion in the lower extremity [23,24].

At the end of the experiment, all mice received ketamine at a dose of 100 mg/kg intraperitoneally and were sacrificed by collecting blood from their abdominal aortas. After the heartbeat and respiration ceased, the mice were monitored for a further 2 min to confirm death.

### 2.1. Histopathologic Assessment

Muscle tissue specimens were immersed in 10% neutral-buffered formalin and fixed for 48 h. Following fixation, tissue specimens were routinely processed and embedded in paraffin. Four µm thick sections were cut from the paraffin blocks using a microtome (Leica RM2245, Nussloch, Germany), and sections were stained with hematoxylin and eosin (H&E) to assess the histologic changes. The H&E-stained muscle sections were observed under 200× and 400× magnifications using a light microscope (Leica DM 4000B, Nussloch, Germany) equipped with a computer, and images were captured using the Leica LAS V4.9 software. Muscle injury was evaluated semi-quantitatively, accounting for the extent of disorganization and degeneration of muscle fibers and inflammatory cell infiltration. Each parameter was scored between 0 and 3 (0, normal; 1, mild; 2, moderate; 3, severe), and then, the degree of IRI was determined using the sum of the scores of these two parameters, corresponding to a value ranging between 0 and 6 [25,26].

### 2.2. Biochemical Assessment

#### 2.2.1. Measurements of TAS/TOS

##### TAS/TOS Measurement

The TAS/TOS test kit (RelAssay Diagnostic^®^, Gaziantep, Turkey) was used according to the manufacturer’s instructions to measure the TAS/TOS levels. The TAS/TOS levels were studied using the method specified in previous publications [27,28].

##### Oxidative Stress Index

The OSI, which has been shown to be an indicator of oxidative stress, is expressed as the ratio of TOS to TAS levels. When calculating the OSI of the samples, the TAS levels were multiplied by 100 to equalize the TOS levels and the units. The results were expressed as arbitrary units (AUs).
$$OSI=\frac{TOS,\;\mu mol\;H_{2}O_{2}\;Equiv./L}{TAS,\;mmol\;Trolox\;Equiv./L\times 100}$$

### 2.3. Paraoxanase (PON) Measurement

Paraoxonase activities were measured spectrophotometrically using commercially available kits (RelAssay Diagnostic^®^, Gaziantep, Turkey).

The rate of paraoxon hydrolysis (diethylpnitrophenylphosphate in 50 mM glycine/NaOH, pH 10.5, containing 1 mM CaCl_2_) was measured by monitoring the increase in absorption at 412 nm at 37 °C. The amount of generated p-nitrophenol was calculated from the molar absorption coefficient at pH 8.5, which was 18.290 M^−1^ cm^−1^ at pH 10.5. One enzyme unit was defined as the amount of enzyme that catalyzed the hydrolysis of 1 μmol of substrate at 37 °C (U/L).

### 2.4. Statistical Analysis

The Statistical Package for the Social Sciences (SPSS, Chicago, IL, USA) 20.0 for Windows was used for statistical analyses. The Shapiro–Wilk test and Q–Q plot test were used to assess the data’s distribution. The results were analyzed using the Kruskal–Wallis test followed by Dunn’s test and one-way ANOVA followed by Tukey’s test. The results were expressed as the mean ± standard deviation (SD) and median (inter quartile range (IQR)). Statistical significance was set at *p* < 0.05.

## 3. Results

### 3.1. Histopathologic Findings

Muscle cells from the control group were observed to display a normal appearance with eosinophilic cytoplasms, polygonal shapes, cross striations, peripherally located nuclei, and forming regular fascicles (Figure 1, group C). In contrast to the control muscle cells, interstitial edema accompanied by inflammatory cell infiltration within the fascicles, and also degenerative changes in muscle fibers such as hypereosinophilic cytoplasms, rounded outlines, and non-peripheral localization of nuclei, and large cytoplasmic defects in some cases, were noted in the specimens of the diabetic mice (Figure 1, group D). The degenerative changes in muscle fibers in the samples of diabetic animals that underwent ischemia-reperfusion were seen to be exacerbated. Varying degrees of degenerative changes such as hyalinization of muscle cells, extensive cytoplasmic vacuolization, and cytoplasmic fragmentation, in which cell integrity was disrupted, were observed in addition to the inflammatory cell infiltration in some areas (Figure 1, groups DIR and DIR-CO).

When the scores for the histopathologic parameters were compared, the disorganization and degeneration of muscle cells, inflammatory cell infiltration, and total injury scores were found to be significantly different between the groups (*p* < 0.001, for all) (Table 1). The degree of disorganization and degeneration of muscle cells in groups DCO, DIR, and DIR-CO were markedly more severe in comparison with group C (*p* = 0.001, *p* < 0.001, and *p* < 0.001, respectively). Additionally, the disorganization and degeneration of muscle cells in group DIR were significantly more prominent than in groups D and DCO (*p* < 0.001 and *p* < 0.001, respectively). The score for disorganization and degeneration of muscle cells in group DIR-CO was markedly higher than in group D, but was significantly lower than that in group DIR (*p* = 0.017 and *p* = 0.002, respectively) (Table 1).

Inflammatory cell infiltration was significantly greater in all other groups compared to group C (*p* = 0.034, *p* = 0.001, *p* < 0.001, and *p* = 0.002, respectively). In addition, the inflammatory cell infiltration score in group DIR was markedly higher than in groups D and DCO (*p* = 0.002 and *p* = 0.033, respectively), whereas it was markedly milder in group DIR-CO in comparison with group DIR (*p* = 0.034) (Table 1).

Finally, the total injury scores of groups D, DCO, DIR, and DIR-CO were found to be higher than in group C (*p* = 0.027, *p* < 0.001, *p* < 0.001, and *p* < 0.001, respectively). Also, the total injury score of group DIR was notably higher than those of groups D and DCO (*p* < 0.001 and *p* < 0.001, respectively). Although the total injury score was markedly higher in group DIR-CO than in group D, it was significantly lower when compared with group DIR (*p* = 0.027 and *p* = 0.001, respectively). According to the total injury score, CeO_2_ appears to alleviate muscle injury (Figure 1, group DIR-CO) (Table 1).

### 3.2. Biochemical Findings

When the groups were compared with each other in terms of muscle tissue TAS levels, there was a significant difference between the groups (*p* = 0.001). TAS levels were found to be significantly lower in all other groups compared to group C (*p* = 0.007, *p* = 0.012, *p* < 0.001, and *p* = 0.002, respectively) (Table 2). In addition, TAS levels were significantly lower in group DIR than in groups D and DCO (*p* = 0.047 and *p* = 0.022, respectively) (Table 2).

When the groups were compared with each other in terms of muscle tissue TOS levels, there was a significant difference between the groups (*p* < 0.001). TOS levels were significantly higher in all other groups compared with group C (*p* = 0.002, *p* = 0.004, *p* < 0.001, and *p* < 0.001, respectively). Similarly, TOS levels were found to be significantly lower in group DIR than in groups D and DCO (*p* < 0.001 and *p* < 0.001, respectively) (Table 2). In group DIRCO, TOS levels were significantly higher than in group DIR (*p* < 0.001) (Table 2).

When the groups were compared with each other in terms of muscle tissue OSIs, there was a significant difference between the groups (*p* < 0.001). OSI levels were found to be significantly higher in all other groups compared to group C (*p* = 0.010, *p* = 0.016, *p* < 0.001, and *p* = 0.001, respectively). Similarly, the OSI was significantly lower in group DIR than in groups D and DCO (*p* < 0.001 and *p* < 0.001, respectively) (Table 2).

When the groups were compared among themselves in terms of muscle tissue paraoxonase (PON) enzyme activity, there was a significant difference between the groups (*p* < 0.001). PON enzyme activity was found to be significantly increased in group DIR compared with groups C, D, and DCO (*p* < 0.001, *p* = 0.002, and *p* < 0.001, respectively). Similarly, PON enzyme activity was significantly increased in group DIRCO compared with group C (*p* = 0.032) (Table 2).

## 4. Discussion

With this study, we wanted to contribute to the limited data in the literature on reducing the effects of ischemia and reperfusion injury on the diabetic group under chronic oxidative stress by evaluating the effects of CeO_2_ use 30 min before IR on muscle tissue in lower extremity IRI in diabetic mice. For this purpose, we examined both histopathologic, oxidative–antioxidative, and lipid peroxidation parameters in muscle tissue in IRI to investigate the effects of CeO_2_ on tissue damage. As a result of our study, we have shown that intraperitoneal CeO_2_ administration in diabetic mice in a lower extremity IRI model significantly reduces oxidative damage in muscle tissue and improves histopathologic findings. This study is one of the first studies to examine the effect of lower extremity ischemia-reperfusion injury on skeletal muscle in diabetic mice.

Tourniquets are frequently used in orthopedic and plastic surgery to create a bloodless surgical field, provide safe and fast surgery, and reduce perioperative blood loss [29]. Lower extremity IRI may occur during the use of a tourniquet, temporary clamping of the aorta in aortic surgery, or acute or bilateral acute femoral artery occlusion [30]. Increasing knowledge of the mechanisms of IRI and developing new treatment strategies to reduce oxidative stress and mitochondrial dysfunction caused by IRI [31] will reduce morbidity by reducing the known distant organ effects of lower extremity IRI [32].

Studies in humans and experimental models indicate that superoxide dismutase 2 (SOD 2), glutathione peroxidase (GPx), and catalase activities in the mitochondria decrease due to hyperglycemia in diabetes. Secondary to the decrease in SOD 2 activity, there is an increase in SOD 1 activity in the cytoplasm. Since catalase and GPx activity decreases, it cannot compensate for the increase in activity in SOD 1, and thus, hydrogen peroxide, which cannot be converted into water, increases in the cell. Accordingly, basal oxidative stress increases in diabetes [3]. It has also been shown that there is mitochondrial dysfunction secondary to hyperglycemia in diabetes [33]. Increased basal oxidative stress and mitochondrial dysfunction in diabetes may cause an increase in the effects of increased ROS and mitochondrial dysfunction due to IRI. Diabetes is a chronic metabolic disease that can lead to macro- and microvascular complications [3]. It is known that diabetes is an important risk factor for peripheral arterial disease (PAD) and it accelerates the progression and increases the severity of the disease [34]. For all these reasons, the effects of ischemia-reperfusion injury are greater in diabetes.

In one study, it was stated that type 1 diabetes worsened IR-induced skeletal muscle injury, and mitochondrial respiration and oxidative stress were more impaired in animals with type 1 diabetes than in non-diabetic animals. This situation became more severe after IRI, and apoptosis and cell damage were more common in animals with diabetes [33].

In our results, the TOS was significantly higher in the diabetic group (group D) than in the control group (group C). TAS was significantly lower in group D than in group C, and OSI was significantly higher in group D than in group C. These results show that oxidative stress is greater in animals with diabetes than in animals without diabetes. This seems to be consistent with our histopathologic results. Disorganization and degeneration of muscle cells were higher in group D than in group C, but the difference was not significant. Inflammatory cell infiltration and total injury score were higher in group D than in group C and the difference was significant. According to these results, inflammatory cell infiltration and total injury scores appear to be more severe in subjects with diabetes.

Mitochondria have important functions such as ATP production, ROS production and detoxification, apoptosis regulation, cytoplasmic and mitochondrial matrix calcium regulation, metabolite synthesis, and catabolism. Dysfunction of mitochondria also plays an important role in the pathogenesis of IRI. Decreased mitochondrial oxidative capacity and dysfunction associated with increased ROS production have also been demonstrated in experimental models of ischemia-reperfusion of the lower extremity using an aortic cross-clamp or leg tourniquet [35].

Oxidative stress occurs as a result of insufficient endogenous catalytic mechanisms in the face of increased production of free radicals, and it causes the progression of the inflammatory process by directly activating the expression of proinflammatory genes and preventing tissue remodeling [36]. We preferred to use TOS, TAS, OSI, and PON1 to analyze the oxidative stress situation. TAS and TOS can provide more accurate results by measuring unknown markers in serum [37]. Because OSI is calculated as the ratio of TOS to TAS, it is a better indicator for systemic oxidative balance [38]. Lipid peroxidation is a pathophysiologic process that occurs as a result of excessive ROS production in IRI and contributes to IRI [39]. PON1 is an enzyme synthesized in the liver that inhibits the oxidation of low-density lipoprotein (LDL) in vitro [40].

Cerium exists in both trivalent (III) and tetravalent (IV) states and can easily switch between these two states; this low-energy change gives CeO_2_ nanoparticles unique catalytic properties. Cerium nanoparticles in the oxide form protect their fluorite structure through oxygen deficiency. In this way, sites for reduction–oxidation reactions are provided. Being able to easily switch between valence states enables CeO_2_ nanoparticles to mimic SOD, CAT, and specific enzyme functions such as phosphatase, oxidase peroxidase, and phosphotriesterase [41]. CeO_2_ nanoparticles have been the subject of many studies with their antioxidant effects and immunomodulatory properties against the overproduction of tissue ROS with enzyme-mimetic effects, mimicking enzymes such as CAT, SOD, and peroxidase [42]. There are few studies in the literature on the effects of CeO_2_ on skeletal muscle ischemia-reperfusion injury in diabetic animals. This study is one of the first studies on skeletal muscle ischemia-reperfusion injury in diabetic animals.

Ozdemirkan et al. [24] examined the effects of desflurane and CeO_2_ on lung tissue following lower extremity limb IRI. In their results, they stated that when CeO_2_ was given in IRI, there were significant reductions in inflammation in histologic findings and in levels of the serum nitric oxide (NO) and malondialdehyde (MDA), which are indicators of oxidative damage. Like this study, Tuncay et al. [18] investigated the efficacy of CeO_2_ on lung tissue in lower extremity IRI in sevoflurane-administered rats, and MDA and NO levels, which are indicators of oxidative stress, were found to be significantly lower in the CeO_2_-applied groups. The authors obtained more positive results in neutrophil infiltration/aggregation and lung tissue total injury scores compared with the IR group.

Ozdemirkan et al. [24] and Tuncay et al. [18] investigated the lung protective effect of 0.5 mg/kg i.p CeO_2_ on lung injury 30 min before lower extremity IRI in rats and observed reduced inflammation histopathologically and oxidative stress biochemically. We applied 0.5 mg/kg i.p CeO_2_ 30 min before ischemia based on studies in the literature. We preferred the intraperitoneal route for the ease of application and rapid absorption [43].

In our study, looking at the OSI, TOS, TAS, and PON1 activities in the muscle tissue, we achieved more positive results in group DCO than in group D. In terms of TAS, we had a positive result in group DIRCO compared with group DIR, and for TOS, OSI, and PON1, we see that the difference in this positive result was significant in group DIRCO compared with group DIR. In the histopathologic evaluation, in terms of disorganization and degeneration of muscle cells, inflammatory cell infiltration and total injury scores in group DIRCO had more positive results than group DIR and the difference was significant. In terms of the same criteria, we had more negative results in group DCO group than in group D, but the difference was not significant. In our study, CeO_2_ administration before IR prevented the increase in oxidative stress and lipid peroxidation markers, and disorganization and degeneration of muscle cells, inflammatory cell infiltration, and total injury score were significantly reduced.

Park et al. [44] injected CeO_2_, prepared at 0.5, 1, and 2 mg/mL, and 300 μL, resulting in CeO_2_ doses of 0.15, 0.3, and 0.6 mg intramuscularly (i.m.) in the left hind limbs of mice with ischemic injury created by ligating the femoral artery, and they monitored tissue reperfusion and hind limb salvage for 3 weeks. In their study, five of seven mice (71.4%) administered 0.6 mg i.m. on day 21 showed no limb loss or necrosis, and it was reported that CeO_2_ had a dose-dependent effect in hind limb salvage. The authors stated that CeO_2_ optimally upregulated the angiogenic growth factor at 0.6 mg and effectively improved endothelial survival by preventing excessive ROS increase in critical limb ischemia.

Byaraa et al. [45] induced an ischemia model by ligating the proximal femoral artery and border vessels of mice. They used CeO_2_-decorated graphene oxide (CeGO) prepared in the form of nano-microlayers from mesenchymal stem cells into a spheroid structure. Compared with the other groups, the highest limb salvage rate was seen in the CeGO-Cell group, with four limb salvages, necrosis in two feet, and one limb loss. The researchers stated that proangiogenic events such as cell sprouting and expression of angiogenic markers (HIF1α, VEGF, FGF2, eNOS) of cellular spheroids increased with CeGO intercalation. In addition, the significantly higher fluorescence signal of ROS was significantly reduced only in the CeGO-Cell group, and they stated that CeGO had an effective role in clearing ROS.

Our study had some limitations. We did not measure hemodynamic parameters such as arterial blood pressure. We could not exclude the effect of hemodynamic changes on oxidative stress. The data we obtained at the end of the experiment are descriptive. The specific mechanism of action was not studied in the experiment. The experiment only includes a 2 h ischemia and 2 h reperfusion period. The effects of CeO_2_ on oxidative stress and inflammation may change during a longer ischemia-reperfusion period. We did not measure levels of markers indicating tissue damage such as creatine kinase, rather, we investigated ROS directly to assess the therapeutic effect of CeO_2_.

## 5. Conclusions

Our results confirm that CeO_2_ has protective effects against IR-induced skeletal muscle injury in diabetic mice. Future studies that will evaluate the mechanisms by which CeO_2_ affects different IR durations will provide us with more in-depth information on the effect of IRI-related tissue damage on the reduction of tissue damage, especially in the diabetic group. Our data shows that prophylactic treatment with CeO_2_ nanoparticles may have a place as a new therapeutic strategy in the treatment of IRI, especially in chronic diseases such as diabetes.

## Figures and Tables

**Figure 1 medicina-60-00752-f001:**
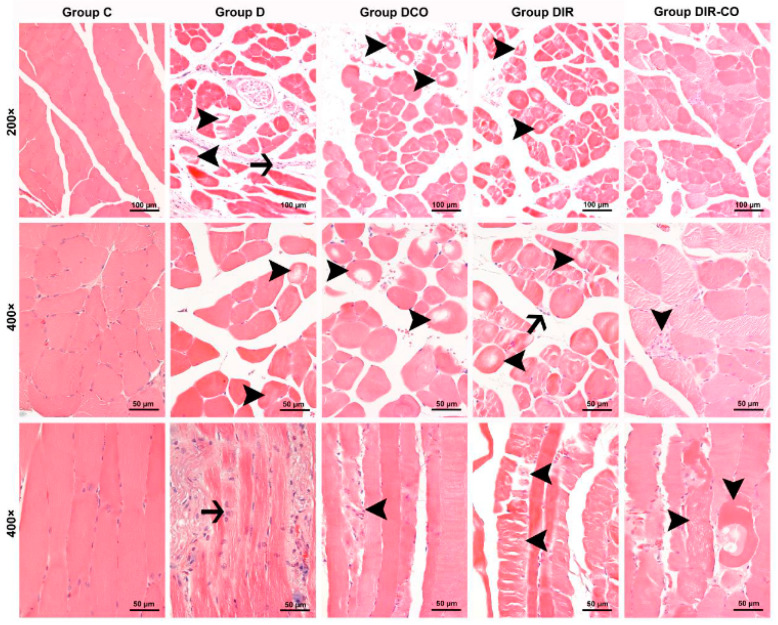
Representative micrographs of H&E-stained skeletal muscle sections captured under 200× and 400× magnifications. Group C, control group; group D, diabetes group; group DCO, diabetes–CeO_2_ group; group DIR, diabetes–ischemia-reperfusion group; group DIR-CO, diabetes–ischemia-reperfusion group administered CeO_2_. Arrowhead, muscle cells display varying degrees of degenerative changes. Arrow, inflammatory cell infiltration. H&E, hematoxylin and eosin staining.

**Table 1 medicina-60-00752-t001:** Histopathologic evaluation and injury scoring data of muscle tissue [median (IQR)].

	Group C (*n* = 6)	Group D (*n* = 8)	Group DCO (*n* = 8)	Group DIR (*n* = 8)	Group DIR-CO (*n* = 8)	*p* **
Disorganization and degeneration of muscle cells	0.00 (0.00–1.00)	1.00 (0.00–2.00)	1.00 (1.00–2.00) *	3.00 (3.–3.00) *,†,‡	2.00 (1.00–2.00) *,†,§	<0.001
Inflammatory cell infiltration	0.00 (0.00–0.00)	0.50 (0.00–1.00) *	1.00 (1.00–1.00) *	1.50 (1.00–2.00) *,†,‡	1.00 (1.00–1.00) *,§	<0.001
Total injury score	0.00 (0.00–1.00)	1.00 (0.00–3.00) *	2.00 (2.00–3.00) *	4.00 (4.00–5.00) *,†,‡	3.00 (2.00–3.00) *,†,§	<0.001

Group C, control group; group D, diabetes group; group DCO, diabetes–CeO_2_ group; group DIR, diabetes–ischemia-reperfusion group; group DIR-CO, diabetes–ischemia-reperfusion group administered CeO_2_. *p* **: Significance value according to the Kruskal–Wallis test, *p* < 0.05. * *p* < 0.05: compared with group C; † *p* < 0.05: compared with group D; ‡ *p* < 0.05: compared with group DCO; § *p* < 0.05: compared with group DIR.

**Table 2 medicina-60-00752-t002:** Biochemical evaluation and oxidant status parameter of muscle tissue [mean ± SD].

	Group C (*n* = 6)	Group D (*n* = 8)	Group DCO (*n* = 8)	Group DIR (*n* = 8)	Group DIR-CO (*n* = 8)	*p* **
TAS (nmol/mL)	1.46 ± 0.47	1.09 ± 0.11 *	1.12 ± 0.22 *	0.81 ± 0.11 *,†,‡	1.03 ± 0.06 *	0.001
TOS (IU/mg·pro)	5.49 ± 2.65	13.99 ± 1.74 *	13.17 ± 3.44 *	28.58 ± 8.03 *,†,‡	16.20 ± 5.13 *,§	<0.001
OSI	3.80 ± 1.18	12.93 ± 1.40 *	11.94 ± 3.45 *	36.09 ± 12.34 *,†,‡	15.87 ± 5.22 *,§	<0.001
PON (IU/mg·pro)	295.57 ± 57.28	355.86 ± 29.48	340.38 ± 57.27	455.86 ± 77.87 *,†,‡	362.57 ± 45.32 *,§	<0.001

*p* **: Significance value according to one-way ANOVA test, *p* < 0.05. * *p* < 0.05: compared with group C; † *p* < 0.05: compared with group D; ‡ *p* < 0.05: compared with group DCO; § *p* < 0.05: compared with group DIR.

## Data Availability

The datasets used and analyzed during the current study are available from the corresponding author on reasonable request.

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
