# Peer review of "The Effect of Cerium Oxide (CeO2) on Ischemia-Reperfusion Injury in Skeletal Muscle in Mice with Streptozocin-Induced Diabetes"

_medicina, 2024, doi:10.3390/medicina60050752_

Round 1
Reviewer 1 Report
Comments and Suggestions for Authors
In the present manuscript entitled “The effect of cerium oxide (CeO2) on ischemia-reperfusion in- 2 jury in skeletal muscle in mice with streptozocin-induced diabetes”. Overall manuscript is well design and has scientific potential for future researcher and clinician. But there are many concerns to the authors to improve the scientific value of the current work which are given below.
Major comments:
1. Authors has to focus on the clinical relevance and novelty of the manuscript.
2. Include more relevant references in discussion to improve the quality of the current manuscript.
3. There are some typo and grammatical errors in the text.
4. Extensive language editing and careful revision is needed.
Recommendation: Major revision
Comments on the Quality of English LanguageNeed to be improve.
Author Response
Re

Reviewer 2 Report
Comments and Suggestions for Authors
Introduction:
The introduction could mention the potential limitations or conflicting evidence in the effectiveness cerium oxide nanoparticles' effectiveness in lower extremity ischemia-reperfusion injury. This will give more depth to the analysis.
A clearer transition to diabetes is needed, it would improve narration.
The connection between diabetes related oxidative stress and the choice cerium oxide nanoparticles as a therapeutic agent should be better stated to improve clarity.
Methods:
How was the assignment of animals to specific groups randomized?
Rationale behind selecting specific time points for assessments, e.g. 4-hour follow-up period, could be more explicitly justified in the context of the study's objectives.
More details on specific statistical tests used for each parameter would improve the paper.
Results & Discussion
A more detailed discussion in the limitation section would serve well. Including small number of animals in each group and the absence of hemodynamic measurements.
Insufficient Discussion of Results:
The results primarily focus on the comparison between groups and the positive outcomes of CeO2 administration. Analyzing unexpected or contradictory results and a discussion of their implications is better. Also exploring any confounding factors that might have influenced the observed outcomes.
Author Response
Reviewer 2
